# Spanish Translation and Cultural Adaptation of the Canadian Assessment of Physical Literacy-2 (CAPL-2) Questionnaires

**DOI:** 10.3390/ijerph19148850

**Published:** 2022-07-21

**Authors:** Raquel Pastor-Cisneros, Jorge Carlos-Vivas, José Carmelo Adsuar, Sabina Barrios-Fernández, Jorge Rojo-Ramos, Alejandro Vega-Muñoz, Nicolás Contreras-Barraza, María Mendoza-Muñoz

**Affiliations:** 1Social Impact and Innovation in Health (InHEALTH) Research Group, Faculty of Sport Sciences, University of Extremadura, 10003 Cáceres, Spain; raquelpc@unex.es (R.P.-C.); sabinabarrios@unex.es (S.B.-F.); jorgerr@unex.es (J.R.-R.); 2Promoting a Healthy Society Research Group (PHeSO), Faculty of Sport Sciences, University of Extremadura, 10003 Cáceres, Spain; jadssal@unex.es; 3Public Policy Observatory, Universidad Autónoma de Chile, Santiago 7500912, Chile; alejandro.vega@uautonoma.cl; 4Facultad de Economía y Negocios, Universidad Andres Bello, Viña del Mar 2531015, Chile; nicolas.contreras@unab.cl; 5Research Group on Physical and Health Literacy and Health-Related Quality of Life (PHYQOL), Faculty of Sport Sciences, University of Extremadura, 10003 Cáceres, Spain; mamendozam@unex.es; 6Departamento de Desporto e Saúde, Escola de Saúde e Desenvolvimento Humano, Universidade de Évora, 7004-516 Évora, Portugal

**Keywords:** assessment, children, cross-cultural adaptation, physical activity, physical education

## Abstract

Background: This study aimed to translate and culturally adapt the questionnaires belonging to the Canadian Assessment of Physical Literacy-2 (CAPL-2) into Spanish and to explore the reliability for its use in children and adolescents aged from 8 to 12 years. Methods: The CAPL-2 questionnaires were translated using the translation–back-translation methodology into Spanish and adapted to the Spanish context. The test–retest reliability and internal consistency of the CAPL-2 questionnaires of this Spanish version were analysed in 57 schoolchildren from a school in the region of Extremadura (Spain). Results: High internal consistency (α = 0.730 to 0.970) and test–retest reliabilities ranging from moderate to almost perfect in the knowledge and understanding domain (ICC = 0.486 to 0.888); from substantial to almost perfect in the motivation and confidence domain (ICC = 0.720 to 0.981); and almost perfect in the daily activity domain (ICC = 0.975) were found. The test–retest correlation was significantly weak to strong (r = 0.266 to 0.815) in both the motivation and confidence and knowledge and understanding domains, except for the third predilection item and the muscular endurance question. Significant test–retest differences were observed in the first intrinsic motivation item (*p* = 0.027) and the knowledge and understanding domain total score (*p* = 0.014). Conclusion: The Spanish version of the CAPL-2 questionnaires, translated and adapted to the context, are reliable measurement tools, serving to complete the full adaptation of the CAPL-2 test battery for use in children aged 8 to 12 years.

## 1. Introduction

Nowadays, childhood obesity is considered one of the main public health problems [1] and stems from several factors, such as physical inactivity and high levels of sedentary lifestyles in young people [2]. It is alarming that 35.3% of the 15–69 years population do not reach the level of physical activity (PA) recommended by the World Health Organization (WHO) [3]. In Spain, many children and young people do not meet the PA guidelines, as they maintain a completely sedentary lifestyle or only engage in very sporadic PA [4], which can lead to short- and long-term health problems [5].

To prevent further exacerbation of this situation, the inclusion of PA programmes in the educational context is important. However, numerous studies has shown that strategies aimed to the promotion of PA in children and young people have not been successful enough [6,7,8]. One of the reasons that may explain these unfavourable results could be the design of most of these programmes, because they are usually focused solely on health improvements, reducing the motivation of adolescents [9]. Furthermore, scientific evidence shows that school-time interventions aimed at increasing PA do not produce significant and relevant long-term changes [10,11].

One possible solution to this issue could be the implementation of strategies and programmes that stimulate the development of children’s capacities for participation in PA as a precursor to increasing PA behaviour itself [12]. In this context, physical literacy (PL) appears, which was defined in the Bulletin of the International Council of Sport Science and Physical Education of the United Nations Educational, Scientific and Cultural Organization (UNESCO) as the motivation, confidence, physical competence, knowledge, and understanding to value and participate in a physically active lifestyle [13]. In other words, it is identified as a multidimensional construct that encompasses all the aforementioned aspects relevant to an individual’s ability to move with competence and attitude in terms of PA [14]. 

Reading, writing, listening, and speaking combine to formulate linguistic literacy, enabling a life of reading and communication. In the same way, physical literacy emerges. Physical literacy is a progressive journey in which the different components (physical competence, daily behaviour, knowledge and understanding, motivation and confidence) interact holistically to facilitate a life of participation and enjoyment of physical activity [15]. Few studies have investigated the association between physical literacy and health, but it has already been shown to be related to body composition, physical fitness, blood pressure, and health-related quality of life (HRQoL) [16].

Recently, physical literacy has caused an increased interest from the scientific community due to its potential to support the development of the whole person in the field of health during childhood [17], as well as the predisposition shown towards PA by children and adolescents in physical education [18].

In this regard, the development of the Canadian Assessment of Physical Literacy-2 (CAPL-2) was one of the most significant initiatives related to childhood obesity in Canada [19]. This assessment is one of the most closely aligned with the concept of child physical literacy, as it assesses daily activity, motivation and confidence, knowledge and understanding, and physical competence. It is a broader alternative to the mere assessment of physical fitness or motor skills. Thus, it provides a robust and comprehensive assessment of physical fitness [20] in 8–12 years children. Its component domains (motivation and confidence, physical competence, knowledge and understanding, and daily behaviour) are assessed through questionnaires and fitness tests [21]. This evaluation has been successfully implemented in different countries such as Australia, Canada, and the United Kingdom [17]. These studies at national and international levels will help to inform political institutions and professionals in this field about the most effective resources and tools to promote PL in children [22].

The correct assessment of PL using the CAPL-2 requires the previous validation of its component tests. In recent years, the CAPL-2 has been translated into several languages such as Danish [12], Greek [23], and Chinese [24]. However, although the CAPL-2 has recently been applied in Spanish children and adolescents [25], there is no study of translation and cultural adaptation into Spanish, nor is there any study that evaluates its reliability. Therefore, the main objective of this study was to translate and culturally adapt the CAPL-2 questionnaires into Spanish, and to explore their reliability for use with 8–12 years children.

## 2. Materials and Methods

### 2.1. Sample Size Power

Post hoc computations for calculating the sample size power were conducted. Thus, regarding an ICC of 0.40 (the minimum ICC obtained) and a 95% confidence interval (CI) for a two-tailed test, a probability error of 0.05, and a sample size of 57 participants, a power of 93% was reported. A 2-way random effects model, single measures, absolute agreement, and ICC were analysed to show the concordance between the test and retest. 

### 2.2. Ethics Approval

The Bioethics and Biosafety Committee at the University of Extremadura approved this study according to the Declaration of Helsinki guidelines (protocol code: 139/2019).

### 2.3. Procedure

The questionnaires belonging to the Canadian Assessment of Physical Literacy-2 (CAPL-2).

The Canadian Assessment of Physical Literacy-2 (CAPL-2) [21] was one of the most significant initiatives related to childhood obesity in Canada [19]. This assessment is one of the most closely aligned to the concept of children’s physical literacy, and it is composed of four domains: (1) daily physical activity behaviour, (2) physical competence, (3) motivation and confidence, and (4) knowledge and understanding. Each domain is composed of different tests, which are summarised in Table 1.

The sections included in the CAPL-2 battery questionnaire belong to 3 of the domains:

Daily activity: the questionnaire includes a question that aims to find out the number of days that the participant was active for at least 60 min. This question will have a score from 1 to 5, with 1 being the lowest possible daily activity and 5 the highest. This score will be added to complete the total score (30 points) of the domain, with the score obtained according to the number of steps recorded during a week by an activity bracelet (25 possible points).

Motivation and confidence: this domain consists of a questionnaire [21] that aims to assess participants’ confidence in the ability to be physically active, and motivation to participate in physical activity. The score is obtained by summing four different dimensions (intrinsic motivation, competition, predilection, and appropriateness). The score for each dimension ranges from 1 to 7.5 points, with the total sum of the domain ranging from 1 to 30 points, where 1 is the worst possible score and 30 the best.

Knowledge and understanding: the total score for this domain comes from an instrument [29] which aims to assess knowledge regarding physical activity. The questionnaire consists of four multiple-choice questions with four answer options (with a score of 1 point each one) and a text where gaps must be filled in to complete a story; in this case, for each correctly filled in gap, one point is assigned, up to a total of 6 points, i.e., the total score will range from 1 to 10 points, with 1 being the lowest possible knowledge of physical activity and 10 the highest.

### 2.4. Translation and Cultural Adaptation

Phase 1: Obtaining the Spanish version of the questionnaires belonging to the Canadian Assessment of Physical Literacy-2 (CAPL-2). Translation and cultural adaptation.

The procedure of the present study followed the methodology used in a translation–back-translation process of questionnaires. For this purpose, the direct and reverse translation methodology was applied following the recommendations of the WHO [30] for the translation and adaptation of instruments, in addition to previous studies conducted on the basis of the CAPL-2 in Danish [12] and Greek [23].

Figure 1 shows the procedure for translation and cultural adaptation of the Spanish version of the questionnaires belonging to the Canadian Physical Literacy Assessment-2. The first phase started with the translation of questionnaires by two Spanish translators who were fluent in English.

The first phase started with the translation of questionnaires by two Spanish translators who were fluent in English. They assessed, from 0 to 10 (0 = no difficulty; 10 = very difficult), the level of difficulty presented when translating each question. After the independent translation process by the two translators of each questionnaire, a consensus meeting was held to obtain a single translation and cultural adaptation of the questionnaires. Thus, the discussion of linguistic differences was carried out, resulting in the first version of the questionnaire. Once the first version was agreed upon, back-translation was carried out. It is based on translating the questionnaire previously translated into Spanish, into the original language (English) by a native English speaker translator who has a good command of the Spanish language. After back-translation, this version was compared with the original (English), resulting in the correctly translated questionnaire.

The Spanish version was evaluated in terms of comprehension in a sample of 10 students, aged between 8 and 12 years old, from an educational centre in Extremadura. These participants underwent a face-to-face interview [31] with the aim of detecting possible mistakes and/or misunderstandings in any of the items, as well as suggestions for improving the questionnaire.

The interview was always conducted by the same person in Spanish. It was based on the following structure: firstly, comprehension was assessed on a three-point ordinal scale: (1) clear and understandable (2) difficult to understand (3) incomprehensible, and then on a numerical scale from 0 to 10, with 0 being very easy to understand and 10 being very difficult to understand. Finally, a paraphrase statement was made, where the respondents expressed in their own words the perceived meaning of the questionnaire items.

The questionnaire was generally evaluated by the participants as clear and understandable, although some minor cultural adaptations were made and agreed upon with the participants. The final version of the questionnaire was obtained after the interviews.

Phase 2: Psychometric properties of the questionnaire.

To analyse the test–retest reliability and internal consistency of the questionnaires, 57 participants were recruited by convenience from a school in the autonomous community of Extremadura (Spain). Once the final versions of the questionnaires were obtained, permission was requested from the school management, and once the study was accepted, parents were asked for permission for their children to participate in the study. The children also signed an informed consent form. Subsequently, each child completed the final questionnaires in their Spanish version on paper, with an interval between the first and second measurement of 21 days. In addition, socio-demographic data such as age, sex, body weight, and height were collected.

### 2.5. Statistical Analyses

All information collected was tabulated into a database specifically designed for this study. The computations were conducted using the Statistical Package for the Social Sciences software (SPSS, version 25.0; IBM, Armonk, NY, USA).

Socio-demographic characteristics (age, sex, height, body weight, fat mass, fat mass percentage, fat-free mass, muscle mass, and body mass index (BMI)) as well as all variables included in computations were described. Data were presented as mean and standard deviation (SD) for both test and retest. Normality and homogeneity of all analysed variables were checked through the Kolmogorov–Smirnov test. All variables showed a non-normal distribution (*p* < 0.05) for test and retest; so non-parametric paired Wilcoxon signed-rank test was applied to analyse systematic differences between the test and retest. Internal consistency and reliability of total domain score, total subdomain scores, and each item’s outcomes were also tested using the Cronbach’s α. Moreover, Spearman’s correlation was used to examine the association between each item and the overall score of every domain and subdomain.

Then, the test–retest reliability or reproducibility was assessed calculating the intraclass correlation coefficient (ICC) with a 95% confident interval (95% CI). A 2-way random effects model, single measures, absolute agreement, and ICC were analysed to show the concordance between the test and retest. ICC values were interpreted following the benchmarks proposed by Landis and Koch [32]: <0.20, slight agreement; 0.21 to 0.40, fair; 0.41 to 0.60, moderate; 0.61 to 0.80, substantial; and >0.80, almost perfect. In addition, standard errors of measurement (SEM) were calculated to measure the range of error of each item, subdomain, and domain. SEM values were calculated from the ICCs and SDs for each session using the higher of the 2 SD measurements to determine the range of error between sessions. SEM was calculated following the equation: SEM = SD × √(1-ICC). The percent error of SEM (SEM%) was also calculated as: SEM% = (SEM/mean) × 100 [33]. Moreover, the minimum detectable changes (MDCs) were calculated as follows [34]: MDC = √2 × 1.96 × SEM. The alpha was set at *p* ≤ 0.05 for all tests.

## 3. Results

### 3.1. Phase 1: Obtaining the Spanish Version of the Questionnaires Belonging to the Canadian Assessment of Physical Literacy-2 (CAPL-2)—Translation and Cultural Adaptation

In the first version of the questionnaire, prior to back-translation, some words, concepts, and terms were modified by consensus, as detailed in Table 2. This version was then back-translated and compared with the original (English) version. After the researchers’ consensus, no significant differences were detected between the two versions.

At the end of this phase, the cognitive interviews reported a good comprehension rating. No comprehension problems were identified by the participants, as all of them rated the questionnaires as clear and understandable, both for the instructions and for all items. An exception occurred in two questionnaire expressions, which were agreed upon and modified, as suggested by them.

After all the modifications mentioned in Table 2, the final version of the questionnaire was obtained.

### 3.2. Phase 2: Psychometric Properties of the Questionnaire

Table 3, Table 4 and Table 5 displays the internal consistency, reproducibility, and systematic differences of the auto-reported physical activity question, motivation and confidence questions, and knowledge and understanding questions from CAPL-2 battery assessment, respectively. Overall, high internal consistency was shown for all questions, domains, and total scores of questionnaires (Cronbach’s α from 0.730 to 0.987), except to the question that referred to the cardiorespiratory fitness definition (Cronbach’s α = 0.656) (Table 3, Table 4 and Table 5). All items at test and retest significantly correlated with the total scores of their domains and subdomains at r > 0.2, except for the 3rd item of predilection from the motivation and confidence domain and the question referring to the muscular endurance definition from the knowledge and understanding domain (Table 3).

Reproducibility outcomes revealed almost perfect test–retest reliability for the self-reported question about physical activity behaviour (ICC = 0.975) and substantial to almost perfect test–retest reliability for each domain and total score of the motivation and confidence domain (ICC = 0.720–0.981); while the test–retest reliability for the knowledge and understanding items and domain was moderate to almost perfect (ICC = 0.486–0.888). The SEM and SEM% values for each item and domain ranged from 0.13 to 0.96 and from 4.11 to 105.13, respectively. The MDC values for each item and domain ranged from 0.35 to 2.65.

Finally, the comparison outcomes between test and retest showed no significant differences in all items and domains (*p* > 0.05), except for the 1st item of the intrinsic motivation subdomain (*p* = 0.027) and the total score of the knowledge and understanding domain (*p* = 0.014).

## 4. Discussion

This study presents the translation and cultural adaptation into Spanish of the questionnaires collected in the CAPL-2. Furthermore, reliability and internal consistency were explored for its application in Spanish children aged from 8 to 12 years. Thus, the aim was to achieve equivalence between the content of the questionnaires of the original CAPL-2 assessment and their subsequent adaptation into Spanish.

The main findings show the success of the translation process of the CAPL-2 questionnaires from English to Spanish, as well as the testing of the face reliability of the CAPL-2 questionnaires for children between 8 and 12 years old in a Spanish population. For a better understanding of the questionnaire, small recurrent changes in the observations produced by the interviewees were included.

The CAPL-2 test battery has been validated in different regions, such as Australia, Canada, and the United Kingdom [17], and has been translated into several languages such as Danish [12], Greek [23], and Chinese [24]. However, although the CAPL-2 has been applied in Spanish children and adolescents [25], it had not been adapted to the context and language of the Spanish population. Therefore, this study is the first that adapts the CAPL-2 questionnaires to the Spanish language and context, testing the internal consistency and the reliability of each of their domains and items.

Regarding the daily activity domain and, more specifically, the self-reported question on weekly PA minutes performed, a high internal consistency was obtained. Moreover, reproducibility results show an almost perfect test–retest reliability, in line with the results reported in the study conducted by Chevance et al. [35] in which they analysed test–retest reliability by studying daily PA behaviours; observing a substantial reliability (ICC > 0.75).

In relation to the motivation and confidence questionnaire [21], the results show a high internal consistency for all subdomains and their respective items; higher than the satisfactory internal consistency found in other studies related to this domain such as the Spanish version of the Sport Motivation Scale (α = 0.70 to 0.80) [36] or motivation towards PE classes (α > 0.70) [37]. Días et al. [38] determined the test–retest reliability and internal consistency of a scale of motivation towards PA in children, also showing a high internal consistency (α = 0.91) [38], as in our study. Reproducibility was reported to have substantial to near perfect test–retest reliability, as in another study with near perfect reliability (ICC > 0.90) [37]. On the other hand, the results showed that all items in the test and retest correlated significantly weakly to strongly with the total score of their domain and subdomain, except the third item of the motivation and confidence domain predilection, probably because this item was more complex for children and adolescents to understand, as it is a multiple-choice question. Significant differences were found in the comparison between the test and the retest of the first item of the subdomain of intrinsic motivation, possibly because the motivation of the children and adolescents increased, as this was the second time they had taken this battery of tests belonging to the CAPL; therefore, their confidence and predisposition towards the assessment increased. The current results are similar to those obtained in a previous study, in which researchers reported that the motivation and confidence questionnaire had good reliability (α = 0.82) [24].

Concerning the knowledge and understanding questionnaire [29], high internal consistency was observed for all items. It is worth mentioning that the test–retest reliability of our study ranged from moderate to almost perfect. Test–retest correlation showed significant associations between all items and the total score of the domain, except to the question referring to the definition of muscular endurance. Significant differences were also found in the test–retest comparison of the knowledge and understanding domain total score, which may be because the questionnaire is not adapted to the context of Spanish children and adolescents, nor to the educational curriculum implemented in our country. However, no studies assessing PA knowledge were found, so we could not establish a comparison between the internal consistency as well as the test–retest reliability and correlation of our study with other similar works. In this sense, further studies are needed to measure reliability as well as internal consistency in the PA knowledge and understanding questionnaire, as there is a clear lack of data, and it corresponds to the domain where the greatest test–retest differences were found.

PL has expanded the possibility of children’s overall development through physical education practices. However, it has lacked valid instruments that measure this concept in a culturally relevant way. Considering the findings of the present study, we can affirm that the CAPL-2 questionnaires, adapted to Spanish, can be considered as reliable instruments to assess the PL of Spanish children and adolescents. Having this instrument adapted to the cultural context of Spain and its population will allow comparisons to be made with other populations in the world with different cultural and socio-economic contexts. It is also important to mention that the entrenchment of PL in the Spanish education system will serve to describe the overall outcome of quality physical education. In addition, this instrument will enable Spanish children and adolescents to participate in structured and comprehensive physical activities. In this sense, another of the great practical implications of this study is that it will promote the adoption and development of a healthy lifestyle in the young generations of our country, thus moving away from sedentary behaviours.

Future studies should also consider translation and cultural adaptation to other languages, with the aim of consolidating a global instrument for the entire world population; even more so, considering the rise of PL. Furthermore, the most successful educational policies and guidelines could be analysed, i.e., those that have been applied by countries whose children and adolescents have the highest PL values. In this way, it would be possible to review the programmes applied in Spain and establish a comparison with those countries with the highest levels of PL in children/adolescents, allowing us to direct our region’s strategies in the same direction.

As limitations of the study, we point out the content used in the domain of knowledge and comprehension, which has a questionnaire adapted to the context of Canadian children and adolescents, as well as to the educational curriculum implemented in that country. Thus, it would be advisable to adapt it to the curriculum used in Spain. The small sample size could also suppose a limitation and explained the weak Cronbach’s (<0.4) alpha for some questionnaires’ items; although a post hoc test revealed a sample size power of 93%. Thus, further research is needed on larger samples that are more representative of Spanish demographic variables, such as anthropometric indices, geographical location, and socio-economic status. Moreover, if the Cronbach’s alpha would be maintained lower than 0.4 in a larger sample size, it should be considered to remove the affected items from the final Spanish version of the battery. Moreover, the fact that this questionnaire has been validated in other languages could give some guarantee that it is useful in Spanish. In this sense, it would be convenient to carry out the validation of the questionnaires, and have the complete battery adapted to Spanish, but of course, taking the present study as a reference.

Future studies, following the methodologies that have been carried out in other regions such as Greece [23], China [24], and Denmark [12], should complete the adaptation of all the tests corresponding to each domain and subdomains of the CAPL-2 battery to Spanish children and adolescents, taking as a starting point the adaptation of the CAPL-2 questionnaires obtained in the present study. Measuring PL achievement longitudinally could be considered as another line of future research, given the dynamic nature of child and adolescent development at these ages. Finally, it would be advisable to explore associations between PL and other concepts that contribute to personal development at these ages (e.g., creativity and empathy). Continuing along these lines, it would be of particular interest to analyse the relationships between the family context, such as the level of physical literacy of the parents and the people who comprise their social environment compared to that of the children themselves.

## 5. Conclusions

Based on the lack of a reliable adaptation of the CAPL-2 questionnaires into Spanish, the first process of translation, cultural adaptation, and reliability of the questionnaires was carried out, and a new instrument for measuring PL in children and adolescents was considered. It is concluded that the Spanish version of the CAPL-2 questionnaires, translated and adapted to the context, is a reliable measurement tool, serving as a starting point to complete the full adaptation of the CAPL-2 test battery for use in children aged from 8 to 12 years.

## Figures and Tables

**Figure 1 ijerph-19-08850-f001:**
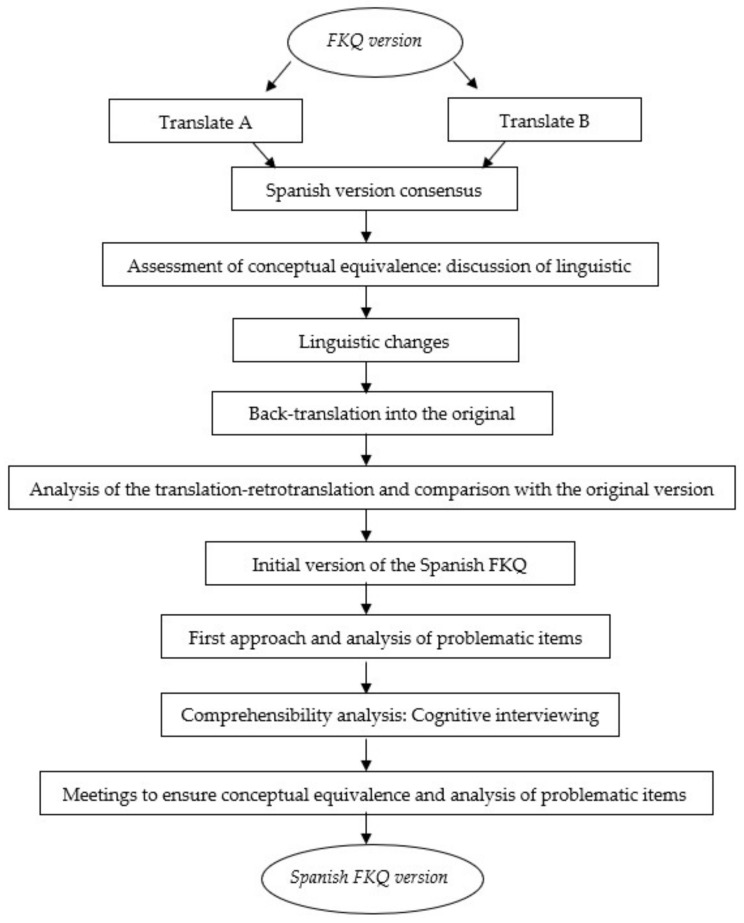
Procedure for translation and cultural adaptation of the Spanish version of the questionnaires belonging to the Canadian Physical Literacy Assessment-2.

**Table 1 ijerph-19-08850-t001:** Tests that make up the Canadian Assessment of Physical Literacy-2 Development (CAPL-2) [21].

Daily physical activity behaviour	Xiaomi mi Band 3 (Xiaomi Corporation, Pekín, China) **Self-reported question: minutes of physical activity performed per week ***
Physical competence	Canadian Agility and Movement Skill Assessment (CAMSA) [26] Plank isometric hold [27] Progressive Aerobic Cardiovascular Endurance Run (PACER) [28]
Motivation and confidence	**1. Canadian Assessment of Physical Literacy-2 Development (CAPL-2) Motivation and Confidence Questionnaires *** [21]
Knowledge and understanding	**1. Canadian Assessment of Physical Literacy-2 Development (CAPL-2) Knowledge and Understanding *** [29]

* The questionnaires described in bold are those selected for translation and cultural adaptation into Spanish.

**Table 2 ijerph-19-08850-t002:** Adaptations made from the initial translated version to the final version of the questionnaire.

English Version (Original)	Agreed Version of the Translations	Adaptations of the Translated Version (First Consensus Version)	Adaptations after Cognitive Interviews
your best guess	la más exacta posible para ti	Tu mejor opción	
There is no time limit, so please take all of the time you need	No hay límite, así que tómate todo el tiempo que necesites	Tómate todo el tiempo que necesites, no hay límite	
What’s Most Like Me?	¿Qué se parece más a mí?	¿qué es más como yo?	
Really true for me/Sort of true for me	Realmente cierto para mi/parcialmente cierto para mi	Muy cierto para mi/poco cierto para mi	Muy verdadero para mi/Poco verdadero para mi
walking, keeping fit, or gym class	andar, mantenerse en forma o ir al gimnasio	Andar o mantenerse en forma	
like soccer, tennis, hockey, dance or swimming	como fútbol, tenis, hockey, baile o natación	como fútbol, tenis, baile o natación	
Count the time you should be active	Calcula el tiempo que deberías estar activo	Que tiempo que deberías estar activo	
what would be the best thing to do?	¿Cuál sería la mejor cosa que podrías hacer?	¿Qué sería lo mejor que podrías hacer?	
the words in the box	las palabras de la caja	las palabras de la caja	las palabras del cuadro
Each word can only be used to fill one blank space in the story	Solo puedes usar una palabra para rellenar el espacio en blanco	Solo puedes usar una palabra para rellenar cada espacio en blanco	

**Table 3 ijerph-19-08850-t003:** Reliability, test–retest and systematic differences of auto-reported physical activity question from CAPL-2 battery assessment.

Physical Activity and Behaviour	Test (*n* = 57)		Retest (*n* = 57)		Reliability Test					
	Mean (SD)	Item-Total Correlation	Mean (SD)	Item-Total Correlation	Cronbach’s α	ICC (95% CI)	*p*-Value †	SEM	%SEM	MDC
Daily behaviour	2.82 (1.58)	N/A	2.88 (1.56)	N/A	0.987	0.975 (0.958–0.985)	0.257	0.25	8.71	0.69

Abbreviations: SD, standard deviation; 95% CI, confidence interval of 95%; ICC, intraclass correlation coefficient; SEM, standard error of measurement; %SEM, standard error of measurement as a percentage; MDC, minimum detectable change; N/A, not applicable; † Wilcoxon signed-rank test. *p*-Values for test-retest comparison in each item. Item-total correlation refers to the magnitude of association between each item with its domain.

**Table 4 ijerph-19-08850-t004:** Reliability, test–retest and systematic differences of motivation and confidence questions from CAPL-2 battery assessment.

Motivation and Confidence	Test (*n* = 57)		Retest (*n* = 57)		Reliability Test					
	Mean (SD)	Item-Total Correlation	Mean (SD)	Item-Total Correlation	Cronbach’s α	ICC (95% CI)	*p*-Value †	SEM	%SEM	MDC
Item 1	2.31 (0.53)	0.564 **	2.23 (0.52)	0.673 **	0.970	0.941 (0.903–0.965)	0.131	0.13	5.60	0.35
Item 2	2.28 (0.58)	0.702 **	2.27 (0.54)	0.656 **	0.951	0.908 (0.849–0.945)	0.680	0.17	7.46	0.47
Item 3	0.99 (0.69)	0.240	1.09 (0.63)	0.342 *	0.906	0.821 (0.712–0.891)	0.152	0.28	26.84	0.78
Predilection	5.58 (0.96)	N/A	5.64 (0.97)	N/A	0.911	0.837 (0.738–0.900)	0.582	0.39	6.94	1.08
Item 1	2.27 (0.54)	0.593 **	2.25 (0.49)	0.444 **	0.944	0.896 (0.829–0.937)	0.524	0.17	7.39	0.46
Item 2	1.04 (0.78)	0.266 *	1.11 (0.76)	0.378 **	0.965	0.930 (0.882–0.958)	0.121	0.20	18.97	0.57
Item 3	2.04 (0.77)	0.649 **	2.01 (0.69)	0.582 **	0.962	0.927 (0.880–0.957)	0.066	0.20	9.71	0.54
Adequacy	5.35 (1.07)	N/A	5.38 (0.94)	N/A	0.926	0.864 (0.779–0.917)	0.597	0.37	6.90	1.03
Item 1	2.16 (0.41)	0.741 **	2.18 (0.41)	0.763 **	0.891	0.805 (0.690–0.880)	0.439	0.18	8.35	0.50
Item 2	2.18 (0.40)	0.651 **	2.18 (0.40)	0.697 **	0.835	0.720 (0.566–0.825)	1.000	0.21	9.61	0.58
Item 3	2.15 (0.44)	0.717 **	2.20 (0.41)	0.718 **	0.939	0.880 (0.803–0.928)	0.058	0.15	6.79	0.41
Intrinsic motivation	6.49 (0.92)	N/A	6.57 (0.91)	N/A	0.905	0.826 (0.722–0.893)	0.297	0.38	5.87	1.06
Item 1	1.85 (0.54)	0.813 **	1.95 (0.46)	0.815 **	0.888	0.786 (0.656–0.870)	**0.027**	0.23	12.21	0.64
Item 2	1.77 (0.60)	0.700 **	1.83 (0.55)	0.736 **	0.942	0.887 (0.814–0.932)	0.090	0.19	10.66	0.53
Item 3	2.10 (0.57)	0.702 **	2.12 (0.48)	0.664 **	0.946	0.898 (0.833–0.939)	0.405	0.17	7.99	0.47
Physical activity Competence	5.72 (1.37)	N/A	5.90 (1.16)	N/A	0.928	0.859 (0.768–0.915)	0.051	0.48	8.19	1.32
Total domain score	23.14 (2.64)	N/A	23.46 (2.69)	N/A	0.933	0.871 (0.788–0.922)	0.079	0.96	4.11	2.65

Abbreviations: SD, standard deviation; 95% CI, confidence interval of 95%; ICC, intraclass correlation coefficient; SEM, standard error of measurement; %SEM, standard error of measurement as a percentage; MDC, minimum detectable change; N/A, not applicable; † Wilcoxon signed-rank test *p*-values for test–retest comparison in each item (significant difference **in bold**); ** *p* < 0.01 for item-total correlation; * *p* < 0.05 for item-total correlation. Item-total correlation refers to the magnitude of association between each item with its domain. Cronbach’s α refers to the value when the item is removed.

**Table 5 ijerph-19-08850-t005:** Reliability, test–retest and systematic differences of knowledge and understanding questions from CAPL-2 battery assessment.

Knowledge and Understanding	Test (*n* = 57)		Retest (*n* = 57)		Reliability Test					
	Mean (SD)	Item-Total Correlation	Mean (SD)	Item-Total Correlation	Cronbach’s α	ICC (95% CI)	*p*-Value †	SEM	%SEM	MDC
Physical activity (PA) guidelines	0.61 (0.49)	0.396 **	0.67 (0.48)	0.292 *	0.943	0.888 (0.816–0.933)	0.083	0.16	25.26	0.45
Cardiorespiratory fitness definition	0.68 (0.47)	0.550 **	0.75 (0.43)	0.450 **	0.656	0.486 (0.262–0.661)	0.248	0.32	45.01	0.90
Muscular endurance definition	0.81 (0.40)	0.335 *	0.86 (0.35)	0.223	0.901	0.814 (0.703–0.886)	0.083	0.16	19.37	0.45
Improve sport skills	0.25 (0.43)	0.312 *	0.32 (0.47)	0.497 **	0.730	0.573 (0.371–0.723)	0.206	0.30	105.13	0.82
PA comprehension	4.49 (0.85)	0.735 **	4.58 (0.86)	0.811 **	0.903	0.821 (0.715–0.890)	0.197	0.36	7.99	1.00
Total domain score	6.84 (1.36)	N/A	7.18 (1.35)	N/A	0.853	0.725 (0.562–0.832)	0.014	0.71	10.14	1.97

Abbreviations: SD, standard deviation; 95% CI, confidence interval of 95%; ICC, intraclass correlation coefficient; SEM, standard error of measurement; %SEM, standard error of measurement as a percentage; MDC, minimum detectable change; N/A, not applicable; † Wilcoxon signed-rank test *p*-values for test–retest comparison in each item; ** *p* < 0.01 for item-total correlation; * *p* < 0.05 for item-total correlation. Item-total correlation refers to the magnitude of association between each item with its domain. Cronbach’s α refers to the value when the item is removed.

## Data Availability

The datasets used during the current study are available from the corresponding authors on reasonable request.

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
