# Peer review of "Spanish Translation and Cultural Adaptation of the Canadian Assessment of Physical Literacy-2 (CAPL-2) Questionnaires"

_ijerph, 2022, doi:10.3390/ijerph19148850_

Round 1

Reviewer 1 Report

Authors have translated the CAPL-2 in Spanish and have tested its psychometric properties. If this study is a first step that is worthy, dealing some important issues could significantly improve the study.  

The first point is the sample size. Authors present post hoc power analysis but post hoc power analysis is always problematic. Sensitivity power analysis should be preferred (see Lakens, 2022). Moreover, even with sensitivity analysis, sample size is really weak for the purpose of the study. Sample size should be at least 300 (several references can be found for the justification of sample size when one wants to validate a scale). I could be open-minded on the fact 300 children is not reached but 56 children is really a small sample size.  

The second point is that the title of the paper highlights the validity but no concurrent nor external validity have been examined. Thus, we only have reliability measures. When researchers want to assess psychometric properties of a scale, the reliability is important but is far from being enough. Authors should provide information about validity (external validity and/or concurrent validity). Moreover, the factor structure is also an important information to ensure the relevance of the translation.

Authors should consider to remove items with an Item-Total Correlation below .4, which is too weak. Note that this low Item-Total Correlation could be explained by the small sample size.

I am not sure to understand how authors compute both item-total correlation and Cronbach’s alpha. Cronbach’s alpha is provided for the whole scale/questionnaire while item-total correlation is computed for each item/measure. Now examine for instance the predilection sub-scale, item 1 has both item-total correlation and Cronbach’s alpha. Thus, I wonder whether Cronbach’s alpha is the value of Cronbach’s alpha when item is removed. This point should be clarified.   

The discussion is redundant with the introduction and with the result section. Redundancies could be removed, and the importance and implications of this scale in the Spanish context could be more developed.

Minor:

-        the spelled-out of ICC is presented line 206 but should be presented line 101.

-        Table 1: no questionnaire is in bold.

-        Line 215, the equation should be SEM%=(SEM/mean) × 100

-        Table 4: no p-value is in bold.

-        Table 4, except if I missed something, the p-value associated to the Wilcoxon test is not clear. Consider removing “for item-total correlation”.

-        Line 337, It is not clear how authors want to “discover relationships between unobserved variables” given that correlation is not causation.

Author Response

Dear reviewer,

The authors' responses are included in the attached pdf.

Kind regards

Author Response

(The authors gave the same response as above.)

Round 2

Reviewer 1 Report

This is a revised version of the manuscript entitled "Spanish translation and cultural adaptation of the Canadian Assessment of Physical Literacy (CAPL-2) Questionnaires.

Authors have corrected what was possible to be corrected from my previous review. If I still believe that the sample size is a limitation, the fact that this questionnaire has been validated in other languages could provide some warranty to ensure that it is useful in Spanish. Authors could perhaps argue on this point in the discussion when they deal with the small sample size.

Moreover, I also have one minor point: Face validity is not convincing evidence for the validation. I urge authors to highlight the reliability and to remove mentions of the validity in the abstract and the discussion.

Author Response

Dear Editor,

Please find attached the document with the authors' response.

Kind regards
